# The Lottery Ticket Hypothesis for Pre-trained BERT Networks

**Tianlong Chen[1], Jonathan Frankle[2], Shiyu Chang[3], Sijia Liu[3], Yang Zhang[3],**
**Zhangyang Wang[1], Michael Carbin[2]**
[1]University of Texas at Austin, [2]MIT CSAIL, [3]MIT-IBM Watson AI Lab, IBM Research
`{tianlong.chen,atlaswang}@utexas.edu`,`{jfrankle,mcarbin}@csail.mit.edu`,
`{shiyu.chang,sijia.liu,yang.zhang2}@ibm.com`

## Abstract

In natural language processing (NLP), enormous pre-trained models like BERT have become the standard starting point for training on a range of downstream tasks, and similar trends are emerging in other areas of deep learning. In parallel, work on the *lottery ticket hypothesis* has shown that models for NLP and computer vision contain smaller *matching* subnetworks capable of training in isolation to full accuracy and transferring to other tasks. In this work, we combine these observations to assess whether such trainable, transferrable subnetworks exist in pre-trained BERT models. For a range of downstream tasks, we indeed find matching subnetworks at 40% to 90% sparsity. We find these subnetworks at (pre-trained) initialization, a deviation from prior NLP research where they emerge only after some amount of training. Subnetworks found on the masked language modeling task (the same task used to pre-train the model) transfer *universally*; those found on other tasks transfer in a limited fashion if at all. As large-scale pre-training becomes an increasingly central paradigm in deep learning, our results demonstrate that the main lottery ticket observations remain relevant in this context. Codes available at `https://github.com/VITA-Group/BERT-Tickets`.

## 1   Introduction

In recent years, the machine learning research community has devoted substantial energy to scaling neural networks to enormous sizes. Parameter-counts are frequently measured in billions rather than millions [1–3], with the time and financial outlay necessary to train these models growing in concert [4]. These trends have been especially pronounced in natural language processing (NLP), where massive BERT models—built on the Transformer architecture [5] and pre-trained in a self-supervised fashion—have become the standard starting point for a variety of downstream tasks [6, 7]. Self-supervised pre-training is also growing in popularity in computer vision [8, 9], suggesting it may again become a standard practice across deep learning as it was in the past [10].

In parallel to this race for ever-larger models, an emerging subfield has explored the prospect of training smaller *subnetworks* in place of the full models without sacrificing performance [11–16]. For example, work on the *lottery ticket hypothesis* (LTH) [16] demonstrated that small-scale networks for computer vision contain sparse, *matching subnetworks* [17] capable of training in isolation from initialization to full accuracy. In other words, we could have trained smaller networks from the start if only we had known which subnetworks to choose. Within the growing body of work on the lottery ticket hypothesis, two key themes have emerged:

**Initialization via pre-training.** In larger-scale settings for computer vision and natural language processing [17–19], the lottery ticket methodology can only find matching subnetworks at an early point in training rather than at random initialization. Prior to this point, these subnetworks perform

no better than those selected by pruning randomly. The phase of training prior to this point can be seen as dense pre-training that creates an initialization amenable to sparsification. This pre-training can even occur using a self-supervised task rather than the supervised downstream task [20, 21].

**Transfer learning.** Finding matching subnetworks with the lottery ticket methodology is expensive. It entails training the unpruned network to completion, pruning unnecessary weights, and *rewinding* the unpruned weights back to their values from an earlier point in training [16]. It is costlier than simply training the full network, and, for best results, it must be repeated many times iteratively. However, the resulting subnetworks transfer between related tasks [22–24]. This property makes it possible to justify this investment by reusing the subnetwork for many different downstream tasks.

These two themes—initialization via pre-training and transfer learning—are also the signature attributes of BERT models: the extraordinary cost of pre-training is amortized by transferring to a range of downstream tasks. As such, BERT models are a particularly interesting setting for studying the existence and nature of trainable, transferable subnetworks. If we treat the pre-trained weights as our initialization, are there matching subnetworks for each downstream task? Do they transfer to other downstream tasks? Are there *universal* subnetworks that can transfer to many tasks with no degradation in performance? Practically speaking, this would allow us to replace a pre-trained BERT with a smaller subnetwork while retaining the capabilities that make it so popular for NLP work.

Although the lottery ticket hypothesis has been evaluated in the context of NLP [18, 19] and transformers [18, 25], it remains poorly understood in the context of pre-trained BERT models.[1] To address this gap in the literature, we investigate how the transformer architecture and the initialization resulting from the lengthy BERT pre-training regime behave in comparison to existing lottery ticket results. We devote particular attention to the transfer behavior of these subnetworks as we search for universal subnetworks that can reduce the cost of fine-tuning on downstream tasks going forward. In the course of this study, we make the following findings:

- Using unstructured magnitude pruning, we find matching subnetworks at between 40% and 90% sparsity in BERT models on standard GLUE and SQuAD downstream tasks.

- Unlike previous work in NLP, we find these subnetworks at (pre-trained) initialization rather after some amount of training. As in previous work, these subnetworks outperform those found by pruning randomly and randomly reinitializing.

- On most downstream tasks, these subnetworks do not transfer to other tasks, meaning that the matching subnetwork sparsity patterns are task-specific.

- Subnetworks at 70% sparsity found using the masked language modeling task (the task used for BERT pre-training) are *universal* and transfer to other tasks while maintaining accuracy.

We conclude that the lottery ticket observations from other computer vision and NLP settings extend to BERT models with a pre-trained initialization. In fact, the biggest caveat of prior work—that, in larger-scale settings, matching subnetworks can only be found early in training—disappears. Moreover, there are indeed universal subnetworks that could replace the full BERT model without inhibiting transfer. As pre-training becomes increasingly central in NLP and other areas of deep learning [8, 9], our results demonstrate that the lottery ticket observations—and the tantalizing possibility that we can train smaller networks from the beginning—hold for the exemplar of this class of learning algorithms.

## 2   Related Work

**Compressing BERT.** A wide range of neural network compression techniques have been applied to BERT models. This includes pruning (in which parts of a model are removed) [27–31], quantization (in which parameters are represented with fewer bits) [32, 33], parameter-sharing (in which the same parameters are used in multiple parts of a model) [34–36], and distilliation (in which a smaller student model is trained to mimic a larger teacher model) [37–45].

We focus on neural network pruning, the kind of compression that was used to develop the lottery ticket hypothesis. In the past decade, computer vision has been the most common application area for

neural network pruning research [46]. Although many ideas from this literature have been applied to Transformer models for NLP, compression ratios are typically lower than in computer vision (e.g., 2x vs. 5x in [25]). Work on pruning BERT models typically focuses on creating small subnetworks after training for faster inference on a specific downstream task. In contrast, we focus on finding compressed models that are *universally* trainable on a range of downstream tasks (a goal shared by [27]). Since we perform a scientific study of the lottery ticket hypothesis rather than an applied effort to gain speedups on a specific platform, we use general-purpose unstructured pruning [47, 25] rather than the various forms of structured pruning common in other work on BERTs [e.g., 30, 31].

**The lottery ticket hypothesis in NLP.** Previous work has found that matching subnetworks exist early in training on Transformers and LSTMs [18, 19] but not at initialization [25]. Concurrent with our research, Prasanna et al. [26] also study the lottery ticket hypothesis for BERT models. Although we share a common topic and application area, our research questions and methods differ, and the results of the two papers are complementary. Prasanna et al. prune entire attention heads and MLP layers in a structured fashion [30], while we prune all parts of the network in an unstructured fashion. Prior work in vision has shown little evidence for the lottery ticket hypothesis when using structured pruning [48]; indeed Prasanna et al. find that all subnetworks ("good" and "bad") have "comparable performance." In contrast, our unstructured pruning experiments show significant performance differences between subnetworks found with magnitude pruning and other baselines (e.g., random pruning). Most importantly, we focus on the question of transferability: do subnetworks found for one task transfer to others, and are there *universal* subnetworks that train well on many tasks? To this end, Prasanna et al. only note the extent to which subnetworks found on different tasks have overlapping sparsity patterns; they do not evaluate transfer performance. Finally, we incorporate nuances of more recent lottery ticket work, e.g., finding matching subnetworks after initialization.

# 3   Preliminaries

In this section, we detail our experimental settings and the techniques we use to identify subnetworks.

**Network.** We use the official BERT model provided by [49, 6] as our starting point for training. We use BERT$_{\text{BASE}}$ [6], which has 12 transformer blocks, hidden state size 768, 12 self-attention heads, and 110M parameters in total. For a particular downstream task, we add a final, task-specific classification layer; this layer contains less than 3% of all parameters in the network [28]. Let $f(x; \theta, \gamma)$ be the output of a BERT neural network with model parameters $\theta \in \mathbb{R}^{d_1}$ and task-specific classification parameters $\gamma \in \mathbb{R}^{d_2}$ on an input example $x$.

**Datasets.** We use standard hyperparameters and evaluation metrics[2] for several downstream NLP tasks as shown in Table 1. All experiment results we presented are calculated from the validation/dev datasets. We divide these tasks into two categories: the self-supervised masked language modeling (MLM) task (which was also used to pre-train the model) [6] and downstream tasks. Downstream tasks include nine tasks from GLUE benchmark [50] and another question-answering dataset, SQuAD v1.1 [51]. Downstream tasks can be divided into three further groups [6]: (a) sentence pair classification, (b) single sentence classification, and (c) question answering.

Table 1: Details of pre-training and fine-tuning. We use standard implementations and hyperparameters [49]. Learning rate decays linearly from initial value to zero. The evaluation metrics are follow standards in [49, 50].

| Dataset | MLM | a.MNLI | a.QQP | a.STS-B | a.WNLI | a.QNLI | a.MRPC | a.RTE | b.SST-2 | b.CoLA | c. SQuAD |
|---|---|---|---|---|---|---|---|---|---|---|---|
| # Train Ex. | 2,500M | 392,704 | 363,872 | 5,760 | 640 | 104,768 | 3,680 | 2,496 | 67,360 | 8,576 | 88,656 |
| # Iters/Epoch | 100,000 | 12,272 | 11,371 | 180 | 20 | 3,274 | 115 | 78 | 2,105 | 268 | 5,541 |
| # Epochs | 0.1 | 3 | 3 | 3 | 3 | 3 | 3 | 3 | 3 | 3 | 2 |
| Batch Size | 16 | 32 | 32 | 32 | 32 | 32 | 32 | 32 | 32 | 32 | 16 |
| Learning Rate | $5\times10^{-5}$ | $2\times10^{-5}$ | $2\times10^{-5}$ | $2\times10^{-5}$ | $2\times10^{-5}$ | $2\times10^{-5}$ | $2\times10^{-5}$ | $2\times10^{-5}$ | $2\times10^{-5}$ | $2\times10^{-5}$ | $3\times10^{-5}$ |
| Optimizer | | | | | AdamW [52] with $\epsilon = 1 \times 10^{-8}$ | | | | | | |
| Eval Metric | Accuracy | Matched Acc. | Accuracy | Pearson Cor. | Accuracy | Accuracy | Accuracy | Accuracy | Accuracy | Matthew's Cor. | F1 |

**Subnetworks.** We study the accuracy when training *subnetworks* of neural networks. For a network $f(x; \theta, \cdot)$, a subnetwork is a network $f(x; m \odot \theta, \cdot)$ with a pruning mask $m \in \{0, 1\}^{d_1}$ (where $\odot$ is the element-wise product). That is, it is a copy of $f(x; \theta, \cdot)$ with some weights fixed to 0.

Let $\mathcal{A}_t^{\mathcal{T}}(f(x;\theta_i,\gamma_i))$ be a training algorithm (e.g., AdamW with hyperparameters) for a task $\mathcal{T}$ (e.g., CoLA) that trains a network $f(x;\theta_i,\gamma_i)$ on task $\mathcal{T}$ for $t$ steps, creating network $f(x;\theta_{i+t},\gamma_{i+t})$. Let $\theta_0$ be the BERT-pre-trained weights. Let $\epsilon^{\mathcal{T}}(f(x;\theta))$ be the evaluation metric of model $f$ on task $\mathcal{T}$.

*Matching subnetwork.* A subnetwork $f(x;m \odot \theta,\gamma)$ is *matching* for an algorithm $\mathcal{A}_t^{\mathcal{T}}$ if training $f(x;m \odot \theta,\gamma)$ with algorithm $\mathcal{A}_t^{\mathcal{T}}$ results in evaluation metric on task $\mathcal{T}$ no lower than training $f(x;\theta_0,\gamma)$ with algorithm $\mathcal{A}_t^{\mathcal{T}}$. In other words:

$$\epsilon^{\mathcal{T}}\left(\mathcal{A}_t^{\mathcal{T}}\left(f\left(x;m \odot \theta,\gamma\right)\right)\right) \geq \epsilon^{\mathcal{T}}\left(\mathcal{A}_t^{\mathcal{T}}\left(f\left(x;\theta_0,\gamma\right)\right)\right)$$

*Winning ticket.* A subnetwork $f(x;m \odot \theta,\gamma)$ is a *winning ticket* for an algorithm $\mathcal{A}_t^{\mathcal{T}}$ if it is a matching subnetwork for $\mathcal{A}_t^{\mathcal{T}}$ and $\theta = \theta_0$.

*Universal subnetwork.* A subnetwork $f(x;m \odot \theta,\gamma_{\mathcal{T}_i})$ is *universal* for tasks $\{\mathcal{T}_i\}_{i=1}^N$ if it is matching for each $\mathcal{A}_{t_i}^{\mathcal{T}_i}$ for appropriate, task-specific configurations of $\gamma_{\mathcal{T}_i}$.

**Identifying subnetworks.** To identify subnetworks $f(x;m \odot \theta,\cdot)$, we use neural network pruning [16, 17]. We determine the pruning mask $m$ by training the unpruned network to completion on a task $\mathcal{T}$ (i.e., using $\mathcal{A}_t^{\mathcal{T}}$) and pruning individual weights with the lowest-magnitudes globally throughout the network [47, 19]. Since our goal is to identify a subnetwork for the pre-trained initialization or for the state of the network early in training, we set the weights of this subnetwork to $\theta_i$ for a specific *rewinding* step $i$ in training. For example, to set the weights of the subnetwork to their values from the pre-trained initialization, we set $\theta = \theta_0$. Previous work has shown that, to find the smallest possible matching subnetworks, it is better to repeat this pruning process iteratively. That is, when we want to find a subnetwork at step $i$ of training:

---

**Algorithm 1** Iterative Magnitude Pruning (IMP) to sparsity $s$ with rewinding step $i$.

---

1: Train the pre-trained network $f(x;\theta_0,\gamma_0)$ to step $i$: $f(x;\theta_i,\gamma_i) = \mathcal{A}_i^{\mathcal{T}}(f(x;\theta_0,\gamma_0))$.
2: Set the initial pruning mask to $m = 1^{d_1}$.
3: **repeat**
4:     Train $f(x;m \odot \theta_i,\gamma_i)$ to step $t$: $f(x;m \odot \theta_t,\gamma_t) = \mathcal{A}_{t-i}^{\mathcal{T}}(f(x;m \odot \theta_i,\gamma_i))$.
5:     Prune 10% of remaining weights [28] of $m \odot \theta_t$ and update $m$ accordingly.
6: **until** the sparsity of $m$ reaches $s$
7: Return $f(x;m \odot \theta_i)$.

---

**Evaluating subnetworks.** To evaluate whether a subnetwork is matching on the original task, we train it using $\mathcal{A}_t^{\mathcal{T}}$ and assess the task-specific performance. To evaluate whether a subnetwork is universal, we train it using several different tasks $\mathcal{A}_{t_i}^{\mathcal{T}_i}$ and assess the task-specific performance.

**Standard pruning.** In some experiments, we compare the size and performance of subnetworks found by IMP to those found by techniques that aim to compress the network after training to reduce inference costs. To do so, we adopt a strategy in which we iteratively prune the 10% of lowest-magnitude weights and train the network for a further $t$ iterations from there (without any rewinding) until we have reached the target sparsity [47, 46, 19]. We refer to this technique as *standard pruning*.

## 4 The Existence of Matching Subnetworks in BERT

In this section, we evaluate the extent to which matching subnetworks exist in the BERT architecture with a standard pre-trained initialization $\theta_0$. In particular, we evaluate four claims about matching subnetworks made by prior work on the lottery ticket hypothesis:

*Claim 1:* In some networks, IMP finds winning tickets $f(x;m_{\text{IMP}} \odot \theta_0,\cdot)$ [16].

*Claim 2:* IMP finds winning tickets at sparsities where randomly pruned subnetworks $f(x;m_{\text{RP}} \odot \theta_i,\cdot)$ and randomly initialized subnetworks $f(x;m_{\text{IMP}} \odot \theta_0',\cdot)$ are not matching [16].

*Claim 3:* In other networks, IMP only finds matching subnetworks $f(x;m_{\text{IMP}} \odot \theta_i,\cdot)$ at some step $i$ *early* in training, or subnetworks initialized at $\theta_i$ outperform those initialized at $\theta_0$ [17].

*Claim 4:* When matching subnetworks are found, they reach the same accuracies at the same sparsities as subnetworks found using standard pruning [19].

Table 2: Performance of subnetworks at the highest sparsity for which IMP finds winning tickets on each task. To account for fluctuations, we consider a subnetwork to be a winning ticket if its performance is within one standard deviation of the unpruned BERT model. Entries with errors are the average across five runs, and errors are the standard deviations. IMP = iterative magnitude pruning; RP = randomly pruning; $\theta_0$ = the pre-trained weights; $\theta_0'$ = random weights; $\theta_0''$ = randomly shuffled pre-trained weights.

| Dataset | MNLI | QQP | STS-B | WNLI | QNLI | MRPC | RTE | SST-2 | CoLA | SQuAD | MLM |
|---|---|---|---|---|---|---|---|---|---|---|---|
| Sparsity | 70% | 90% | 50% | 90% | 70% | 50% | 60% | 60% | 50% | 40% | 70% |
| Full BERT$_{\text{BASE}}$ | $82.4 \pm 0.5$ | $90.2 \pm 0.5$ | $88.4 \pm 0.3$ | $54.9 \pm 1.2$ | $89.1 \pm 1.0$ | $85.2 \pm 0.1$ | $66.2 \pm 3.6$ | $92.1 \pm 0.1$ | $54.5 \pm 0.4$ | $88.1 \pm 0.6$ | $63.5 \pm 0.1$ |
| $f(x, m_{\text{IMP}} \odot \theta_0)$ | $82.6 \pm 0.2$ | $90.0 \pm 0.2$ | $88.2 \pm 0.2$ | $54.9 \pm 1.2$ | $88.9 \pm 0.4$ | $84.9 \pm 0.4$ | $66.0 \pm 2.4$ | $91.9 \pm 0.5$ | $53.8 \pm 0.9$ | $87.7 \pm 0.5$ | $63.2 \pm 0.3$ |
| $f(x, m_{\text{RP}} \odot \theta_0)$ | 67.5 | 76.3 | 21.0 | 53.5 | 61.9 | 69.6 | 56.0 | 83.1 | 9.6 | 31.8 | 32.3 |
| $f(x, m_{\text{IMP}} \odot \theta_0')$ | 61.0 | 77.0 | 9.2 | 53.5 | 60.5 | 68.4 | 54.5 | 80.2 | 0.0 | 18.6 | 14.4 |
| $f(x, m_{\text{IMP}} \odot \theta_0'')$ | 70.1 | 79.2 | 19.6 | 53.3 | 62.0 | 69.6 | 52.7 | 82.6 | 4.0 | 24.2 | 42.3 |

**Claim 1: Are there winning tickets?** To study this question, we (1) run IMP on a downstream task $\mathcal{T}$ to obtain a sparsity pattern $m_{\text{IMP}}^{\mathcal{T}}$ and (2) initialize the resulting subnetwork to $\theta_0$. This produces a subnetwork $f(x; m_{\text{IMP}}^{\mathcal{T}} \odot \theta_0, \cdot)$ that we can train on task $\mathcal{T}$ to evaluate whether it is a winning ticket. This experiment is identical to the lottery ticket procedure proposed by Frankle & Carbin [16].

We indeed find winning tickets for the MLM task and all downstream tasks (Table 2). To account for fluctuations in performance, we consider a subnetwork to be a winning ticket if the performance of full BERT is within one standard deviation of the performance of the subnetwork.[3] The highest sparsities at which we find these winning tickets range from 40% (SQuAD) and 50% (MRPC and CoLA) to 90% (QQP and WNLI). There is no discernible relationship between the sparsities for each task and properties of the task itself (e.g., training set size).[4]

**Claim 2: Are IMP winning tickets sparser than randomly pruned or initialized subnetworks?** Prior work describes winning tickets as a "combination of weights and connections capable of learning" [16]. That is, both the specific pruned weights and the specific initialization are necessary for a winning ticket to achieve this performance. To assess these claims in the context of BERT, we train a subnetwork $f(x; m_{\text{RP}} \odot \theta_0, \cdot)$ with a random pruning mask (which evaluates the importance of the pruning mask $m_{\text{IMP}}$) and a subnetwork $f(x; m_{\text{IMP}} \odot \theta_0', \cdot)$ with a random initialization (which evaluates the importance of the pre-trained initialization $\theta_0$). Table 2 shows that, in both cases, performance is far lower than that of the winning tickets; for example, it drops by 15 percentage points on MNLI when randomly pruning and 21 percentage points when reinitializing. This confirms the importance of the specific pruned weights and initialization in this setting.

Since the pre-trained BERT weights are our initialization, the notion of sampling a new random initialization is less precise than when there is an explicit initialization distribution. We therefore explore another form of random initialization: shuffling the BERT pre-trained weights within each layer to obtain a new initialization $\theta_0''$ [20]. In nearly all cases, training from $\theta_0''$ outperforms training from $\theta_0'$, although it still falls far short of the performance of the winning tickets. This suggests that the per-layer weight distributions from BERT pre-training are a somewhat informative starting point.

The random pruning and random reinitialization experiments in Table 2 are at the highest sparsities for which we find winning tickets using IMP. In Figure 1, we compare IMP and randomly pruned subnetworks across all sparsities for CoLA, SST-2, and SQuAD. The accuracy of random pruning is close to that of IMP only at the lowest sparsities (10% to 20%) if at all, confirming that the structure of the pruning mask is crucial for the performance of the IMP subnetworks at high sparsities.

**Claim 3: Does rewinding improve performance?** It is noteworthy that we find winning tickets at *non-trivial* sparsities (i.e., sparsities where random pruning cannot find winning tickets). The only other settings where this has previously been observed are small networks for MNIST and CIFAR-10 [16]. Winning tickets have not been found in larger-scale settings, including transformers [25, 18]

Table 3: Performance of subnetworks found using IMP with rewinding to the steps in the left column and standard pruning (where subnetworks are trained using the final weights from the end of training).

| Dataset | MNLI | QQP | STS-B | WNLI | QNLI | MRPC | RTE | SST-2 | CoLA | SQuAD | MLM |
|---|---|---|---|---|---|---|---|---|---|---|---|
| Sparsity | 70% | 90% | 50% | 90% | 70% | 50% | 60% | 60% | 50% | 40% | 70% |
| Full BERT$_{BASE}$ | 82.39 | 90.19 | 88.44 | 54.93 | 89.14 | 85.23 | 66.16 | 92.12 | 54.51 | 88.06 | 63.48 |
| Rewind 0% (i.e., $\theta_0$) | 82.45 | 89.20 | 88.12 | 54.93 | 88.05 | 84.07 | 66.06 | 91.74 | 52.05 | 87.74 | 63.07 |
| Rewind 5% | 82.99 | 88.98 | 88.05 | 54.93 | 88.85 | 83.82 | 62.09 | 92.43 | 53.38 | 87.78 | 63.18 |
| Rewind 10% | 82.93 | 89.08 | 88.11 | 54.93 | 89.02 | 84.07 | 62.09 | 92.66 | 52.61 | 87.77 | 63.49 |
| Rewind 20% | 83.08 | 89.21 | 88.28 | 55.75 | 88.87 | 85.78 | 61.73 | 92.89 | 52.02 | 87.36 | 63.82 |
| Rewind 50% | 82.94 | 89.54 | 88.41 | 53.32 | 88.72 | 85.54 | 62.45 | 92.66 | 52.20 | 87.26 | 64.21 |
| Standard Pruning | 82.11 | 89.97 | 88.51 | 52.82 | 89.88 | 85.78 | 62.95 | 90.02 | 52.00 | 87.12 | 63.77 |

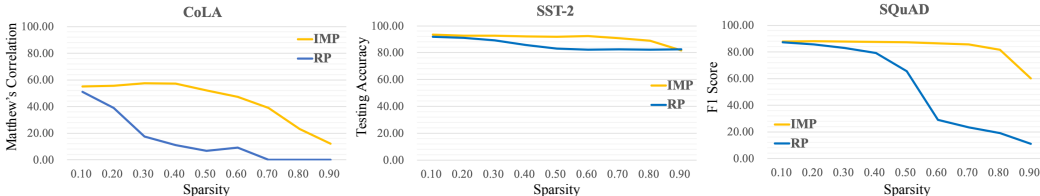

Figure 1: Comparing IMP and random pruning across sparsities on CoLA, SST-2, and SQuAD.

and LSTMs [18, 19] for NLP tasks. The existence of winning tickets here implies that the BERT pre-trained initialization has different properties than other NLP settings with random initializations.

In settings where winning tickets have not been found, IMP still finds matching subnetworks at non-trivial sparsities. However, these subnetworks must be initialized to the state of the network $\theta_i$ after $i$ steps of training [17, 22, 18] rather than to initialization $\theta_0$. This procedure is known as *rewinding*, since the subnetwork found by IMP is *rewound* to the weights at iteration $i$ rather than *reset* to initialization $\theta_0$. Although we have already found winning tickets, we still evaluate the possibility that rewinding improves performance. Table 3 shows the error when rewinding to iteration $i$, where $i$ is 5%, 10%, 20%, and 50% of the way through training.[5] For example, on SQuAD v1.1, $i \in \{1000, 2000, 4000, 10000\}$.

Rewinding does not notably improve performance for any downstream task. In fact, in some cases (STS-B and RTE), performance drops so much that the subnetworks are no longer matching, even with our two percentage point margin. This is a notable departure from prior work where rewinding had, at worst, no effect on accuracy. A possible explanation for the particularly poor results on STS-B and RTE is that their small training sets result in overfitting.

**Claim 4: Do IMP subnetworks match the performance of standard pruning?** In the limit of rewinding, we can initialize the IMP subnetwork to weights $\theta_t$ at the end of training, producing a subnetwork $f(x; m \odot \theta_t, \cdot)$. Doing so is the basis of a standard way of pruning neural networks: train the network to completion, prune, and train further to recover performance without rewinding [47]. Renda et al. [19] show that, in other settings, IMP subnetworks rewound early in training reach the same accuracies at the same sparsities as subnetworks found by this standard pruning procedure. In Table 3, we see that results vary depending on the task. For some tasks (QQP, QNLI, MRPC, MLM), standard pruning improves upon winning ticket performance by up to two percentage points. For others (STS-B, WNLI, RTE, SST-2), performance drops by up to three percentage points. The largest drops again occur for tasks with small training sets where standard pruning may overfit.

**Summary.** We evaluated the extent to which prior lottery ticket observations manifest in the context of pre-trained BERT models. We confirmed that the standard observations hold: there are indeed matching subnetworks at non-trivial sparsities, and the sparsity patterns and initializations of these subnetworks matter for achieving this performance. The most notable departure from prior work is that we find winning tickets at (pre-trained) initialization at non-trivial sparsities, a return to the original lottery ticket paradigm [16] that was previously observed only in small-scale vision settings.

| Subnetworks on the Source Tasks (Sparsity %) | MNLI | QQP | STS-B | WNLI | QNLI | MRPC | RTE | SST-2 | CoLA | SQuAD v1.1 | MLM | |
|---|---|---|---|---|---|---|---|---|---|---|---|---|
| MNLI (70%) | 82.56 | 89.20 | 84.77 | 47.89 | 87.34 | 72.14 | 60.75 | 90.83 | 11.19 | 82.90 | 57.52 | 2 |
| QQP (70%) | 80.87 | 89.95 | 84.18 | 52.11 | 87.27 | 72.30 | 60.17 | 88.80 | 16.50 | 81.57 | 57.64 | 1 |
| STS-B (70%) | 80.05 | 88.26 | 87.34 | 56.34 | 86.17 | 72.71 | 57.40 | 87.92 | 4.31 | 80.74 | 57.59 | 1 |
| WNLI (70%) | 79.70 | 87.52 | 67.13 | 53.87 | 84.96 | 69.90 | 55.23 | 87.27 | 0.00 | 80.31 | 57.75 | 1 |
| QNLI (70%) | 80.80 | 88.75 | 83.16 | 54.93 | 88.89 | 71.73 | 58.96 | 89.56 | 3.65 | 82.44 | 57.47 | 3 |
| MRPC (70%) | 79.98 | 87.88 | 81.25 | 56.34 | 85.66 | 75.57 | 54.87 | 88.15 | 7.48 | 79.89 | 57.74 | 1 |
| RTE (70%) | 80.18 | 88.18 | 79.50 | 55.87 | 86.49 | 71.57 | 58.37 | 88.15 | 1.55 | 80.77 | 57.78 | 1 |
| SST-2 (70%) | 80.15 | 88.44 | 77.61 | 53.99 | 85.77 | 70.67 | 56.92 | 89.99 | 7.52 | 81.05 | 57.76 | 2 |
| CoLA (70%) | 80.06 | 88.29 | 77.48 | 54.93 | 86.30 | 70.83 | 55.60 | 88.57 | 38.89 | 81.01 | 57.81 | 1 |
| SQuAD v1.1 (70%) | 80.90 | 88.90 | 84.09 | 53.99 | 89.40 | 72.06 | 59.93 | 90.18 | 8.03 | 86.37 | 57.47 | 3 |
| (IMP) MLM (70%) | 82.59 | 90.03 | 87.43 | 55.05 | 89.44 | 81.58 | 59.81 | 91.86 | 47.15 | 86.54 | 63.16 | 6 |
| Pruning $\theta_0$ (70%) | 82.46 | 89.62 | 85.28 | 53.52 | 89.13 | 72.55 | 58.84 | 91.06 | 32.21 | 85.33 | 57.09 | 4 |

Transfer Tasks

Figure 2: Transfer Winning Tickets. The performance of transferring IMP subnetworks between tasks. Each row is a source task $\mathcal{S}$. Each column is a target task $\mathcal{T}$. Each cell is $\textsc{Transfer}(\mathcal{S}, \mathcal{T})$: the performance of finding an IMP subnetwork at 70% sparsity on task $\mathcal{S}$ and training it on task $\mathcal{T}$ (averaged over three runs). **Dark cells mean the IMP subnetwork on task $\mathcal{S}$ is a winning ticket on task $\mathcal{T}$ at 70% sparsity, i.e., $\textsc{Transfer}(\mathcal{S}, \mathcal{T})$ is within one standard deviation of the performance of the full BERT network.** The number on the right is the number of target tasks $\mathcal{T}$ for which transfer performance is at least as high as same-task performance. The last row is the performance when the pruning mask comes from directly pruning the pre-trained weights $\theta_0$.

## 5    Transfer Learning for BERT Winning Tickets

In Section 4, we found that there exist winning tickets for BERT models using the pre-trained initialization. In this section, we investigate the extent to which IMP subnetworks found for one task transfer to other tasks. We have reason to believe that such transfer is possible: Morcos et al. [22] and Mehta [23] show that matching subnetworks transfer between vision tasks.[6] To investigate this possibility in our setting, we ask three questions:

*Question 1:* Are winning tickets $f(x; m_{\text{IMP}}^{\mathcal{S}} \odot \theta_0, \cdot)$ for a *source task* $\mathcal{S}$ also winning tickets for other *target tasks* $\mathcal{T}$?

*Question 2:* Are there patterns in the transferability of winning tickets? For example, do winning tickets transfer better from tasks with larger training sets [22] or tasks that are similar?

*Question 3:* Does initializing subnetworks with the pre-trained initialization $\theta_0$ transfer better than initializing from weights that have been trained on a specific task $\mathcal{T}$ (i.e., using rewinding)?

**Question 1: Do winning tickets transfer?** To study this behavior, we first identify a subnetwork $f(x, m_{\text{IMP}}^{\mathcal{S}} \odot \theta_0, \cdot)$ on a source task $\mathcal{S}$, following the same routine as in Section 4. We then train it on all other target tasks $\{\mathcal{T}_i\}$ and evaluate its performance. When we perform transfer, we sample a new, randomly initialized classification layer for the specific target task. Let $\textsc{Transfer}(\mathcal{S}, \mathcal{T})$ be the performance of this procedure with source task $\mathcal{S}$ and target task $\mathcal{T}$.

One challenge in analyzing transfer is that the winning tickets from Section 4 are different sizes. This introduces a confounding factor: larger winning tickets (e.g., 40% sparsity for CoLA) may perform disproportionately well when transferring to tasks with smaller winning tickets (e.g., 90% sparsity for

| Subnetworks on the Source Tasks (Sparsity %) | MNLI | QQP | STS-B | WNLI | QNLI | MRPC | RTE | SST-2 | CoLA | SQuAD v1.1 | MLM | |
|---|---|---|---|---|---|---|---|---|---|---|---|---|
| MNLI (70%) | 0.00 | -0.75 | -2.57 | -5.99 | -1.55 | -3.44 | 2.38 | 0.84 | -27.70 | -3.46 | -5.63 | 2 |
| QQP (70%) | -1.69 | 0.00 | -3.16 | -1.76 | -1.62 | -3.27 | 1.80 | -1.19 | -22.39 | -4.80 | -5.52 | 1 |
| STS-B (70%) | -2.51 | -1.69 | 0.00 | 2.47 | -2.72 | -2.86 | -0.97 | -2.07 | -34.58 | -5.62 | -5.57 | 1 |
| WNLI (70%) | -2.86 | -2.42 | -20.21 | 0.00 | -3.93 | -5.67 | -3.13 | -2.72 | -38.89 | -6.06 | -5.40 | 0 |
| QNLI (70%) | -1.76 | -1.20 | -4.18 | 1.06 | 0.00 | -3.84 | 0.60 | -0.42 | -35.24 | -3.93 | -5.68 | 2 |
| MRPC (70%) | -2.58 | -2.07 | -6.09 | 2.47 | -3.23 | 0.00 | -3.50 | -1.84 | -31.41 | -6.47 | -5.41 | 1 |
| RTE (70%) | -2.38 | -1.77 | -7.84 | 2.00 | -2.40 | -4.01 | 0.00 | -1.84 | -37.34 | -5.60 | -5.37 | 1 |
| SST-2 (70%) | -2.42 | -1.50 | -9.73 | 0.12 | -3.12 | -4.90 | -1.45 | 0.00 | -31.36 | -5.31 | -5.39 | 1 |
| CoLA (70%) | -2.50 | -1.65 | -9.86 | 1.06 | -2.59 | -4.74 | -2.77 | -1.42 | 0.00 | -5.36 | -5.34 | 1 |
| SQuAD v1.1 (70%) | -1.66 | -1.05 | -3.25 | 0.12 | 0.51 | -3.52 | 1.56 | 0.19 | -30.86 | 0.00 | -5.69 | 4 |
| (IMP) MLM (70%) | 0.02 | 0.09 | 0.09 | 1.18 | 0.55 | 6.01 | 1.44 | 1.87 | 8.27 | 0.17 | 0.00 | 10 |
| Pruning $\theta_0$ (70%) | -0.10 | -0.33 | -2.06 | -0.35 | 0.24 | -3.02 | 0.47 | 1.07 | -6.68 | -1.04 | -6.07 | 3 |

Transfer Tasks

Figure 3: Transfer vs. Same-Task. The performance of transferring IMP subnetworks between tasks. Each row is a source task $\mathcal{S}$. Each column is a target task $\mathcal{T}$. Each cell is $\text{TRANSFER}(\mathcal{S}, \mathcal{T}) - \text{TRANSFER}(\mathcal{T}, \mathcal{T})$: the transfer performance at 70% sparsity minus the same-task performance at 70% sparsity (averaged over three runs). **Dark cells mean the IMP subnetwork found on task $\mathcal{S}$ performs at least as well on task $\mathcal{T}$ as the subnetwork found on task $\mathcal{T}$.**

QQP). We therefore prune IMP subnetworks to fixed sparsity 70%; this means that, on some tasks, these subnetworks are too sparse to be winning tickets (i.e., not within one standard deviation of unpruned performance). In Appendix S2, we show the same transfer experiment for other sparsities.

Each cell in Figure 2 shows the transfer performance $\text{TRANSFER}(\mathcal{S}, \mathcal{T})$ and marked as Dark when it surpasses the performance of training the unpruned BERT on task $\mathcal{T}$. This determines whether the subnetwork from task $\mathcal{S}$ is a winning ticket for task $\mathcal{T}$. However, many subnetworks are too sparse to be winning tickets on the task for which they were found. To account for this behavior, we evaluate whether transfer was successful by comparing *same-task performance* $\text{TRANSFER}(\mathcal{T}, \mathcal{T})$ (i.e., how a subnetwork found for target task $\mathcal{T}$ performs on task $\mathcal{T}$) and transfer performance $\text{TRANSFER}(\mathcal{S}, \mathcal{T})$, and report the performance difference (i.e., $\text{TRANSFER}(\mathcal{S}, \mathcal{T}) - \text{TRANSFER}(\mathcal{T}, \mathcal{T})$) in Figure 3. Dark cells mean transfer performance matches or exceeds same-task performance.

Although there are cases where transfer performance matches same-task performance, they are the exception. Of the eleven tasks we consider, subnetworks from only three source tasks transfer to more than two other tasks. However, if we permit a drop in transfer performance by 2.5 percentage points, then seven source tasks transfer to at least half of the other tasks.

The MLM task produces the subnetwork with the best transfer performance. It is universal in the sense that transfer performance matches same-task performance in all cases.[7] It is unsurprising that these subnetworks have the best transfer performance, since MLM is the task that was used to create the pre-trained initialization $\theta_0$. Note that we obtain this subnetwork by running IMP on the MLM task: we iteratively train from $\theta_0$ on the MLM task, prune, rewind to $\theta_0$, and repeat. If we directly prune the pre-trained weights $\theta_0$ without this iterative process (last row of Figure 2 and 3), performance is worse: transfer performance matches same-task performance in only four cases. With that said, these results are still better than any source task except MLM.

| SQuAD v1.1 (70 %) | MNLI | QQP | STS-B | WNLI | QNLI | MRPC | RTE | SST-2 | CoLA | SQuAD v1.1 | MLM | |
|---|---|---|---|---|---|---|---|---|---|---|---|---|
| Rewind to 5% | 0.44 | -0.03 | 0.40 | -9.86 | 0.25 | 4.16 | 3.25 | 0.11 | 2.35 | -0.03 | 0.01 | 8 |
| Rewind to 10% | 0.32 | 0.02 | 0.55 | -5.63 | 0.11 | 4.90 | 3.61 | 0.23 | 0.93 | -0.18 | -0.01 | 8 |
| Rewind to 20% | 0.21 | -0.21 | 0.55 | 2.82 | 0.22 | 4.88 | 5.41 | 0.11 | 1.21 | -0.32 | -0.04 | 8 |
| Rewind to 50% | 0.12 | -0.26 | 0.46 | 0.00 | -0.18 | 5.39 | 5.05 | 0.00 | 1.79 | -0.46 | -0.03 | 7 |
| Standard Pruning | -0.23 | -0.14 | -0.23 | 0.00 | -0.14 | 3.18 | 2.16 | -0.46 | -6.56 | -0.44 | -0.86 | 3 |

Figure 4: Transfer performance for subnetworks found using IMP with rewinding and standard pruning on SQuAD. Each cell shows the relative performance compared to rewinding to $\theta_0$.

**Question 2: Are there patterns in subnetwork transferability?** Transferability seems to correlate with the number of training examples for the task. MRPC and WNLI have among the smallest training sets; the MRPC subnetwork transfers to just one other task, and the WNLI subnetwork does not transfer to any others. On the other hand, MNLI and SQuAD have among the largest training sets, and the resulting subnetworks transfer to four and three other tasks, respectively. MLM, which has by far the largest training set, also produces the subnetwork that transfers best. Interestingly, we do not see any evidence that transfer is related to task type (using the groupings described in Table 1).

**Question 3: Does initializing to $\theta_0$ lead to better transfer?** One possible argument in favor of using winning tickets for transfer is that their initialization is more "general" than weights that have been trained on a specific downstream task $\mathcal{S}$. To evaluate this hypothesis, we study transferring subnetworks that have been rewound to $\theta_i$ (rather than $\theta_0$) and subnetworks found using standard pruning. We focus on SQuAD, which had the second best transfer performance after MLM.[8] In nearly all cases, rewinding leads to the same or higher performance on target tasks, and standard pruning also has little detrimental effect on transfer performance. This suggests that, at least for SQuAD, weights trained on the source task seem to *improve* transfer performance rather than degrade it.

**Summary.** We evaluated the transferability of IMP subnetworks. Transfer was most effective for tasks with larger training sets and particularly MLM (which resulted in a universal subnetwork). Pruning the pre-trained weights directly resulted in transfer performance as good as using the IMP subnetwork for the best downstream task (SQuAD). Finally, transferring from $\theta_0$ was not noticeably better than transferring from weights fine-tuned on SQuAD by various amounts.

# 6 Conclusions and Implications

We investigated the lottery ticket hypothesis in the context of pre-trained BERT models. We found that the main lottery ticket observations continue to hold: using the pre-trained initialization, BERT contains sparse subnetworks at non-trivial sparsities that can train in isolation to full performance on a range of downstream tasks. Moreover, there are universal subnetworks that transfer to all of these downstream tasks. This transfer means we can replace the full BERT model with a smaller subnetwork while maintaining its signature ability to transfer to other tasks.

In this sense, our results can be seen as a possible second stage of the BERT pre-training process: after the initial pre-training, perform IMP using MLM to arrive at an equally-capable subnetwork with far fewer parameters. In future work, we would be interested to examine the speedup results on a hardware platform for the training and/or inference phases of our method. For example, in the range of 70%-90% unstructured sparsity, XNNPACK [53] has already shown significant speedups over dense baselines on smartphone processors.

## Acknowledgements

We would like to express our deepest gratitude to the MIT-IBM Watson AI Lab, in particular John Cohn for generously providing us with the computing resources necessary to conduct this research.

Wang's work is in part supported by the NSF Energy, Power, Control, and Networks (EPCN) program (Award number: 1934755), and by an IBM faculty research award.

## Broader Impact

We do not believe that this research poses significant risk of societal harm. This research is scientific in nature, exploring the behavior of an empirical phenomenon (the lottery ticket hypothesis) in a new setting. In general, lottery ticket subnetworks appear to have the same expressive power as the full networks from which they originate, meaning we have not enabled new learning paradigms that were not already possible with the full BERT model. The largest potential societal impact of the research is that, on the appropriate hardware platforms, it may be possible to reduce the cost (both energy costs and financial costs) of fine-tuning BERT models on downstream tasks by using our universal subnetworks rather than the full BERT network.

## Footnotes

[1]A concurrent study by Prasanna et al. [26] also examines the lottery ticket hypothesis for BERTs. However, there are important differences in the questions we consider and our results. See Section 2 for a full comparison.

[2]For MNLI, QQP, STS-B and MRPC, we report the other evaluation metrics in Appendix A.

[3]In Appendix A, we show the same data for the highest sparsities where IMP subnetworks exactly match or surpass the error of the unpruned BERT on each task. In addition, the mean, median and best performance of five independent runs are also presented.

[4]In Appendix D, we compare the overlap in sparsity patterns found for each of these tasks in a manner similar to Prasanna et al. We find that subnetworks for downstream tasks share a large fraction of pruned weights in common, while there are larger differences for subnetworks for the MLM task.

[5]After rewinding, each subnetwork $f(x; m \odot \theta_i, \cdot)$ is then trained for the full $t$ iterations.

[6]As further evidence, IMP pruning masks for downstream tasks are similar to one another (Appendix D).

[7]In Appendix B, we study the universality of MLM subnetworks in a broader sense. Let the highest sparsity at which we can find a winning ticket for task $\mathcal{T}$ be $s\%$. We find that, for five of the downstream tasks, the MLM subnetwork at sparsity $s\%$ is also a winning ticket for $\mathcal{T}$. For a further three tasks, the gap is within half a percentage point. The gap remains small (1.6 and 2.6 percentage points) for the two remaining tasks.

[8]Although MLM transferred best, rewinding should not significantly affect these results. $\theta_0$ is the result of running MLM for 1M steps, so rewinding to $\theta_i$ is equivalent to pre-training for 1M+$i$ steps. In Appendix B, we show the effect of rewinding on transfer for MLM and MNLI.

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
