[Supplementary Material]

# A  Further Results on the Existence of Matching Subnetworks in BERT

In Table 2 in Section 3, we show the highest sparsities for which IMP subnetwork performance is within one standard deviation of the unpruned BERT model on each task. In Table 4 below, we plot the same information for the highest sparsities at which IMP subnetworks match or exceed the performance of the unpruned BERT model on each task. The sparsest winning tickets are in many cases larger under this stricter criterion. QQP goes from 90% sparsity to 70% sparsity, STS-B goes from 50% sparsity to 40% sparsity, QNLI goes from 70% sparsity to 50% sparsity, MRPC goes from 50% sparsity to 40% sparsity, RTE goes from 60% sparsity to 50%, SST-2 goes from 60% sparsity to 50%, CoLA goes from 50% sparsity to 40% sparsity, SQuAD goes from 40% sparsity to 20% sparsity, and MLM goes from 70% sparsity to 50% sparsity. MNLI and WNLI are unchanged.

As broader context for the relationship between sparsity and accuracy, Figure 11 shows the performance of IMP subnetworks across all sparsities on each task.

Table 4: Performance of subnetworks at the highest sparsity for which IMP finds winning tickets (performance matches or exceeds that of the full BERT network) on each task. Entries with errors are the average across five runs, and errors are standard deviations.

| Dataset | MNLI | QQP | STS-B | WNLI | QNLI | MRPC | RTE | SST-2 | CoLA | SQuAD v1.1 | MLM |
|---|---|---|---|---|---|---|---|---|---|---|---|
| Sparsity Level | 70% | 70% | 40% | 90% | 50% | 50% | 50% | 50% | 50% | 20% | 50% |
| Full BERT$_{BASE}$ | $82.4 \pm 0.5$ | $90.2 \pm 1.1$ | $88.4 \pm 0.6$ | $54.9 \pm 1.2$ | $89.1 \pm 1.6$ | $85.2 \pm 0.1$ | $66.2 \pm 4.4$ | $92.1 \pm 0.3$ | $54.5 \pm 1.1$ | $88.1 \pm 1.0$ | $63.5 \pm 0.2$ |
| $f(x; m_{\text{IMP}} \odot \theta_0)$ | $82.6 \pm 0.2$ | $90.3 \pm 0.3$ | $88.4 \pm 0.3$ | $54.9 \pm 1.2$ | $89.8 \pm 0.2$ | $85.5 \pm 0.1$ | $66.2 \pm 2.9$ | $92.2 \pm 0.3$ | $57.3 \pm 0.1$ | $88.2 \pm 0.2$ | $63.6 \pm 0.2$ |
| $f(x; m_{\text{RP}} \odot \theta_0)$ | 67.5 | 78.1 | 30.4 | 53.5 | 77.4 | 71.3 | 51.0 | 83.1 | 12.4 | 85.7 | 49.6 |
| $f(x; m_{\text{IMP}} \odot \theta_0^s)$ | 61.0 | 77.8 | 15.3 | 53.5 | 60.8 | 68.4 | 54.5 | 80.2 | 0.0 | 19.0 | 14.7 |
| $f(x; m_{\text{IMP}} \odot \theta_0^s)$ | 70.1 | 85.5 | 26.6 | 53.3 | 79.6 | 68.4 | 52.7 | 82.6 | 6.02 | 77.4 | 47.9 |

In Table 5, we show the mean, median and best performance of five runs for Full BERT$_{BASE}$ and our found sparse subnetworks. Our best numbers in Table 5 are in line with those reported by HuggingFace [49].

Table 5: Performance of subnetworks at the highest sparsity for which IMP finds winning tickets on each task. We consider a subnetwork to be a winning ticket when the performance of the unpruned BERT model is within one standard deviation the subnetwork's performance. Entries with errors are the average across five runs, and errors are the standard deviations. The **mean, median and best** performance of five runs are collected.

| Dataset | MNLI | QQP | STS-B | WNLI | QNLI | MRPC | RTE | SST-2 | CoLA | SQuAD | MLM |
|---|---|---|---|---|---|---|---|---|---|---|---|
| Eval Metric | Matched Acc. | Accuracy | Pearson Cor. | Accuracy | Accuracy | Accuracy | Accuracy | Accuracy | Matthew's Cor. | F1 | Accuracy |
| Sparsity | 70% | 90% | 50% | 90% | 70% | 50% | 60% | 60% | 50% | 40% | 70% |
| Full BERT$_{BASE}$ (mean ± stddev) | $82.4 \pm 0.5$ | $90.2 \pm 0.5$ | $88.4 \pm 0.3$ | $54.9 \pm 1.2$ | $89.1 \pm 1.0$ | $85.2 \pm 0.1$ | $66.2 \pm 3.6$ | $92.1 \pm 0.1$ | $54.5 \pm 0.4$ | $88.1 \pm 0.6$ | $63.5 \pm 0.1$ |
| $f(x, m_{\text{IMP}} \odot \theta_0)$ (mean ± stddev) | $82.6 \pm 0.2$ | $90.0 \pm 0.2$ | $88.2 \pm 0.2$ | $54.9 \pm 1.2$ | $88.9 \pm 0.4$ | $84.9 \pm 0.4$ | $66.0 \pm 2.4$ | $91.9 \pm 0.5$ | $53.8 \pm 0.9$ | $87.7 \pm 0.5$ | $63.2 \pm 0.3$ |
| Full BERT$_{BASE}$ (median) | 82.4 | 90.3 | 88.5 | 54.9 | 89.2 | 85.2 | 67.0 | 92.1 | 54.4 | 88.2 | — |
| $f(x, m_{\text{IMP}} \odot \theta_0)$ (median) | 82.6 | 90.0 | 88.3 | 54.9 | 88.8 | 85.0 | 66.5 | 92.0 | 53.7 | 87.9 | — |
| Full BERT$_{BASE}$ (best) | 83.6 | 90.9 | 88.7 | 56.3 | 90.5 | 85.3 | 69.3 | 92.3 | 56.1 | 88.9 | — |
| $f(x, m_{\text{IMP}} \odot \theta_0)$ (best) | 82.9 | 90.3 | 88.5 | 56.3 | 89.5 | 85.4 | 69.0 | 92.6 | 55.1 | 88.4 | — |
| HuggingFace Reported | 83.95 | 90.72 | 89.70 | 53.52 | 89.04 | 83.82 | 61.01 | 93.00 | 57.29 | 88.54 | — |

In Table 6, we report both common evaluation metrics for MNLI, QQP, STS-B, and MRPC datasets. Besides STS-B (50% Pearson vs. 40% Spearman), winning ticket sparsities are the same on these tasks regardless of the metric.

Table 6: Both standard evaluation metrics for MNLI, QQP, STS-B, and MRPC at the highest sparsity for which IMP finds winning tickets using the corresponding metric on each task. We consider a subnetwork to be a winning ticket when the performance of the unpruned BERT model is within one standard deviation the subnetwork's performance. Entries with errors are the average across five runs, and errors are the standard deviations.

| Dataset | MNLI | | QQP | | STS-B | | MRPC | |
|---|---|---|---|---|---|---|---|---|
| Eval Metric | Matched Acc. | Mismatched Acc. | Accuracy | F1 | Pearson Cor. | Spearman Cor. | F1 | Accuracy |
| Sparsity | 70% | 70% | 90% | 90% | 50% | 40% | 50% | 50% |
| Full BERT$_{BASE}$ (mean ± stddev) | $82.4 \pm 0.5$ | $83.2 \pm 0.4$ | $90.2 \pm 0.5$ | $87.0 \pm 0.3$ | $88.4 \pm 0.3$ | $88.2 \pm 0.3$ | $89.2 \pm 0.2$ | $85.2 \pm 0.1$ |
| $f(x, m_{\text{IMP}} \odot \theta_0)$ (mean ± stddev) | $82.6 \pm 0.2$ | $83.1 \pm 0.1$ | $90.0 \pm 0.2$ | $86.9 \pm 0.1$ | $88.2 \pm 0.2$ | $88.0 \pm 0.2$ | $89.0 \pm 0.3$ | $84.9 \pm 0.4$ |

Table 7: Comparison between the performance of subnetworks found on the corresponding target task and the transfer performance of subnetworks found on MLM task, at the same sparsity level.

| Target Task | MNLI | QQP | STS-B | WNLI | QNLI | MRPC | RTE | SST-2 | CoLA | SQuAD |
|---|---|---|---|---|---|---|---|---|---|---|
| Sparsity | 70% | 90% | 50% | 90% | 70% | 50% | 60% | 60% | 50% | 40% |
| Full BERT$_{\text{BASE}}$ | $82.4 \pm 0.5$ | $90.2 \pm 0.5$ | $88.4 \pm 0.3$ | $54.9 \pm 1.2$ | $89.1 \pm 1.0$ | $85.2 \pm 0.1$ | $66.2 \pm 3.6$ | $92.1 \pm 0.1$ | $54.5 \pm 0.4$ | $88.1 \pm 0.6$ |
| $f(x; m_{\text{IMP}}^{\mathcal{T}} \odot \theta_0)$ | $82.6 \pm 0.2$ | $90.0 \pm 0.2$ | $88.2 \pm 0.2$ | $54.9 \pm 1.2$ | $88.9 \pm 0.4$ | $84.9 \pm 0.4$ | $66.0 \pm 2.4$ | $91.9 \pm 0.5$ | $53.8 \pm 0.9$ | $87.7 \pm 0.5$ |
| $f(x; m_{\text{IMP}}^{\text{MLM}} \odot \theta_0)$ | $82.6 \pm 0.2$ | $88.5 \pm 0.2$ | $88.0 \pm 0.2$ | $56.3 \pm 1.0$ | $89.4 \pm 0.3$ | $83.4 \pm 0.2$ | $65.1 \pm 3.1$ | $92.0 \pm 0.4$ | $51.6 \pm 0.3$ | $87.8 \pm 0.1$ |

# B  Further Results on Transfer Learning for BERT Winning Tickets

**Universality.** In Section 4, we showed that the MLM subnetworks at 70% sparsity are universal in the sense that transfer performance $\text{TRANSFER}(\mathcal{S}, \mathcal{T})$ is at least as high as same-task performance $\text{TRANSFER}(\mathcal{T}, \mathcal{T})$. In Table 7, we investigate universality in a broader sense. Let the highest sparsity at which we can find a winning ticket for task $\mathcal{T}$ be $s\%$. We study whether the MLM subnetwork at sparsity $s\%$ is also a winning ticket for $\mathcal{T}$. For five of the ten downstream tasks, this is the case. On a further three tasks, the gap is half a percentage point or less. Only on MRPC (1.6 percentage point gap) and CoLA (2.6 percentage point gap) are the gaps larger. We conclude that IMP subnetworks found using MLM are universal in a broader sense: they winning tickets or nearly so at the most extreme sparsities for which we can find winning tickets on each downstream task.

**Additional sparsities.** In Figure 5, we show the equivalent results to Figure 2 in the main body of the paper at 50% sparsity rather than 70% sparsity. Figures 7 and 8 present the transfer performance of subnetworks found using IMP with rewinding and standard pruning on MLM and MNLI, respectively. Our results suggest that weights trained on the source task seem to improve transfer performance for MNLI, while degrading it for MLM.

# C  Finding Subnetworks with Multi-task Pruning

In Figure 9, we study IMP on networks trained with a multi-task objective. We observe that IMP subnetworks of BERT trained on the combination of the MLM task and downstream tasks have a marginal transfer performance gain compared with the one found on the single MLM task. This suggests that we cannot significantly improve on the transfer performance of the MLM task by incorporating information from other downstream tasks before pruning.

# D  Similarity between Sparsity Patterns

In Figure 10, we compare the overlap in sparsity patterns found for each of these tasks in a manner similar to Prasanna et al. Each cell contains the relative overlap ratio (i.e., $\frac{m_i \cap m_j}{m_i \cup m_j}\%$) between masks (i.e., $m_i, m_j$) from task $\mathcal{T}_i$ and task $\mathcal{T}_j$. We find that subnetworks for downstream tasks are remarkably similar: they share more than 90% of pruned weights in common, while there are larger differences for subnetworks for the MLM task. The similarity between downstream tasks makes it surprising that the subnetworks transfer so poorly between tasks. Likewise, the lack of similarity to the MLM task makes it surprising that this subnetwork transfers so well.

# E  Influence of Training Dataset Size on Transfer

In Section 5, we observed that IMP subnetworks for the MLM task had the best transfer performance of all tasks at 70% sparsity. One possible reason for this performance is the sheer size of the training dataset, which has more than six times as many training examples as the largest downstream task. To study the effect of the size of the training set, we study the transfer performance of IMP subnetworks for the MLM task when we artificially constrain the size of the training dataset. In particular, we reduce the size of the training dataset to match the size of SST-2 (67,360 training examples), CoLA (8,576 training examples), and SQuAD (88,656 training examples). We also consider using 160,000 training examples—the number of examples that MLM will use during a single iteration of IMP.

In Table 8 contains the results of these experiments. We observe that MLM subnetworks found with more training samples have a small but consistent transfer performance improvement. However,

| Subnetworks on the Source Tasks (Sparsity %) | MNLI | QQP | STS-B | WNLI | QNLI | MRPC | RTE | SST-2 | CoLA | SQuAD v1.1 | MLM | |
|---|---|---|---|---|---|---|---|---|---|---|---|---|
| MNLI (50%) | -0.42 | 0.78 | -0.47 | -16.90 | 1.38 | -2.20 | 1.09 | 0.00 | -5.68 | -1.27 | -1.10 | 6 |
| QQP (50%) | 1.44 | 0.78 | -0.43 | -19.72 | 1.21 | -3.92 | -4.69 | -0.80 | -3.77 | -1.93 | -1.05 | 4 |
| STS-B (50%) | 1.21 | 0.74 | -0.18 | -9.86 | 0.86 | -1.96 | -3.24 | -0.68 | -4.54 | -1.92 | -1.02 | 5 |
| WNLI (50%) | 0.86 | 0.73 | -2.27 | -16.90 | 0.76 | -6.61 | -2.52 | -0.46 | -1.42 | -1.89 | -0.95 | 5 |
| QNLI (50%) | 1.22 | 0.69 | -1.03 | -18.31 | 0.65 | -2.69 | -6.49 | -0.11 | -0.90 | -1.52 | -0.96 | 4 |
| MRPC (50%) | 0.83 | 0.74 | -2.14 | -25.35 | 0.50 | -0.22 | -2.52 | -0.57 | -1.96 | -1.97 | -0.97 | 3 |
| RTE (50%) | 1.06 | 0.78 | -2.67 | -15.49 | 1.07 | -2.20 | -0.88 | 0.00 | -2.21 | -1.73 | -1.00 | 6 |
| SST-2 (50%) | 1.22 | 0.72 | -2.75 | -22.54 | 1.12 | -5.63 | -6.13 | -2.39 | -3.01 | -1.64 | -0.98 | 3 |
| CoLA (50%) | 1.29 | 0.79 | -2.42 | -21.13 | 0.86 | -5.39 | -8.66 | -0.34 | -0.71 | -1.81 | -0.98 | 5 |
| SQuAD v1.1 (50%) | 1.36 | 0.95 | -2.19 | -19.72 | 1.58 | -5.63 | -4.33 | -0.34 | -1.58 | -1.10 | -1.11 | 4 |
| (IMP) MLM (50%) | 1.00 | 0.80 | -0.47 | -0.03 | 0.85 | -1.09 | -0.80 | 0.35 | -2.91 | -0.78 | 0.12 | 8 |
| Pruning $\theta_0$ (50%) | 0.86 | 0.70 | -1.02 | 1.41 | 1.29 | -8.82 | -7.22 | 0.12 | -3.05 | -1.18 | -1.00 | 5 |

Figure 5: The performance of transferring IMP subnetworks between tasks. Each row is a source task $\mathcal{S}$. Each column is a target task $\mathcal{T}$. Let $\textsc{Transfer}(\mathcal{S}, \mathcal{T})$ be the performance of finding an IMP subnetwork at $50\%$ sparsity on task $\mathcal{S}$ and training it on task $\mathcal{T}$. Each cell is $\textsc{Transfer}(\mathcal{S}, \mathcal{T})$ minus the performance of the full network $f(x; \theta_0)$ on task $\mathcal{T}$. Dark cells mean transfer performance $\textsc{Transfer}(\mathcal{S}, \mathcal{T})$ is at least as high as same-task performance $\textsc{Transfer}(\mathcal{T}, \mathcal{T})$; light cells mean it is lower. The number on the right is the number of target tasks $\mathcal{T}$ for which transfer performance is at least as high as same-task performance. The last row is the performance when the pruning mask comes from directly pruning the pre-trained weights $\theta_0$.

transfer performance still matches or outperforms same-task performance even when the number of MLM training examples is constrained to be the same as the downstream training set size.

Table 8: Transfer performance of MLM subnetworks $f(x; m_{\text{IMP}}^{\text{MLM}} \odot \theta_0)$ obtained from different number of training examples. Here we study the subnetwork at the 70% sparsity level.

| Subnetwork | # Train Examples | SST-2 | CoLA | SQuAD v1.1 |
|---|---|---|---|---|
| $f(x; m_{\text{IMP}}^{\mathcal{T}} \odot \theta_0)$ (70%) | 67,360 | 90.9 | - | - |
| | 8,576 | - | 39.4 | - |
| | 88,656 | - | - | 86.4 |
| $f(x; m_{\text{IMP}}^{\text{MLM}} \odot \theta_0)$ (70%) | 67,360 | 91.4 | - | - |
| | 8,576 | - | 44.2 | - |
| | 88,656 | - | - | 86.1 |
| $f(x; m_{\text{IMP}}^{\text{MLM}} \odot \theta_0)$ (70%) | 160,000 | 91.9 | 46.7 | 86.5 |

| Subnetworks on the Source Tasks (Sparsity %) | MNLI | QQP | STS-B | WNLI | QNLI | MRPC | RTE | SST-2 | CoLA | SQuAD v1.1 | Pre-training | |
|---|---|---|---|---|---|---|---|---|---|---|---|---|
| MNLI (70%) | 82.56 | 89.20 | 84.77 | 47.89 | 87.34 | 72.14 | 60.75 | 90.83 | 11.19 | 82.90 | 57.52 | 2 |
| QQP (90%) | 77.00 | 90.04 | 69.78 | 56.34 | 82.68 | 69.61 | 55.96 | 83.83 | 0.00 | 36.30 | 49.21 | 2 |
| STS-B (50%) | 83.60 | 90.23 | 88.23 | 45.07 | 90.00 | 83.33 | 62.82 | 91.06 | 48.97 | 86.24 | 63.36 | 7 |
| WNLI (90%) | 60.73 | 80.49 | 6.61 | 54.93 | 61.98 | 68.38 | 52.35 | 80.11 | 0.00 | 16.29 | 23.21 | 1 |
| QNLI (70%) | 80.80 | 88.75 | 83.16 | 54.93 | 88.89 | 71.73 | 58.96 | 89.56 | 3.65 | 82.44 | 57.47 | 3 |
| MRPC (50%) | 83.22 | 90.23 | 86.83 | 29.58 | 89.64 | 84.90 | 63.54 | 91.17 | 51.55 | 86.19 | 63.41 | 7 |
| RTE (60%) | 82.50 | 89.68 | 85.69 | 53.52 | 89.04 | 75.00 | 66.06 | 90.60 | 37.99 | 83.94 | 61.20 | 4 |
| SST-2 (60%) | 82.90 | 89.52 | 85.39 | 46.48 | 88.93 | 74.51 | 59.21 | 91.91 | 41.00 | 84.00 | 61.09 | 4 |
| CoLA (50%) | 83.68 | 90.28 | 86.55 | 33.80 | 90.00 | 79.90 | 57.40 | 91.40 | 53.75 | 86.35 | 63.40 | 6 |
| SQuAD v1.1 (40%) | 83.73 | 90.55 | 88.30 | 36.62 | 91.45 | 80.64 | 66.06 | 92.66 | 53.42 | 87.70 | 64.20 | 9 |
| (IMP) MLM (70%) | 82.59 | 90.03 | 87.43 | 55.05 | 89.44 | 81.58 | 59.81 | 91.86 | 47.15 | 86.54 | 63.16 | 6 |
| Pruning $\theta_0$ (70%) | 82.46 | 89.62 | 85.28 | 53.52 | 89.13 | 72.55 | 58.84 | 91.06 | 32.21 | 85.33 | 57.09 | 4 |

Transfer Tasks

Figure 6: The performance of transferring IMP subnetworks between tasks. Each row is a source task $\mathcal{S}$. Each column is a target task $\mathcal{T}$. Each cell is TRANSFER$(\mathcal{S}, \mathcal{T})$: the performance of finding an IMP subnetwork at the sparsity where we find a winning ticket on task $\mathcal{S}$ and training it on task $\mathcal{T}$ (averaged over three runs). **Dark cells mean the IMP subnetwork on task $\mathcal{S}$ is a winning ticket on task $\mathcal{T}$ at the corresponding sparsity, i.e., TRANSFER$(\mathcal{S}, \mathcal{T})$ is within one standard deviation of the performance of the full BERT network.**

| MLM (70 %) | MNLI | QQP | STS-B | WNLI | QNLI | MRPC | RTE | SST-2 | CoLA | SQuAD v1.1 | MLM | |
|---|---|---|---|---|---|---|---|---|---|---|---|---|
| Rewind to 5% | 0.21 | -0.29 | -3.22 | -1.41 | 0.44 | -4.42 | 0.58 | -1.38 | -4.39 | -0.45 | 0.11 | 4 |
| Rewind to 10% | 0.18 | -0.15 | -3.14 | -1.41 | -0.07 | -3.68 | -3.75 | -1.26 | -5.14 | -0.19 | 0.42 | 2 |
| Rewind to 20% | -0.20 | -0.22 | -3.60 | -2.82 | 0.37 | -3.44 | -0.50 | -1.03 | -1.21 | -0.03 | 0.75 | 2 |
| Rewind to 50% | -0.43 | -0.15 | -2.81 | -11.27 | -0.03 | -1.72 | -4.83 | -1.26 | -4.66 | -0.02 | 1.14 | 1 |
| Standard Pruning | 0.10 | -0.33 | -2.99 | -1.41 | 0.20 | -4.17 | -0.50 | -1.49 | -2.17 | -0.26 | 0.70 | 3 |

Figure 7: Transfer performance for subnetworks found using IMP with rewinding and standard pruning on MLM. Each cell shows the relative performance compared to rewinding to $\theta_0$.

| MNLI (70 %) | MNLI | QQP | STS-B | WNLI | QNLI | MRPC | RTE | SST-2 | CoLA | SQuAD v1.1 | MLM | |
|---|---|---|---|---|---|---|---|---|---|---|---|---|
| Rewind to 5% | 0.54 | -0.20 | 1.36 | 0.00 | -0.36 | -0.38 | 1.80 | 1.14 | 7.91 | 0.46 | -0.01 | 7 |
| Rewind to 10% | 0.48 | -0.12 | 1.65 | 0.00 | -0.23 | 1.83 | 1.44 | 1.49 | 4.67 | 0.82 | -0.03 | 8 |
| Rewind to 20% | 0.63 | -0.03 | 1.68 | 1.41 | 0.30 | 0.60 | 2.16 | 1.60 | 7.00 | 0.93 | -0.01 | 9 |
| Rewind to 50% | 0.49 | 0.01 | 1.99 | -1.41 | 0.58 | 1.09 | 2.89 | 1.37 | 10.81 | 1.03 | -0.07 | 9 |
| Standard Pruning | -0.34 | -0.12 | 0.01 | -15.49 | -0.14 | -0.13 | 3.61 | 0.91 | 8.28 | -0.22 | -1.62 | 4 |

Figure 8: Transfer performance for subnetworks found using IMP with rewinding and standard pruning on MNLI. Each cell shows the relative performance compared to rewinding to $\theta_0$.

Figure 9: Transfer performance for subnetworks, at 70% sparsity level, found using IMP with multiple tasks.

Figure 10: The overlap in sparsity patterns found on each tasks.

Figure 11: Performance of subnetworks found using IMP across sparsities on each task.