[Reviews · NeurIPS 2020]

Review 1

Summary and Contributions: This paper studies an interesting and important problem in the current BERT-driven NLP field: Does the lottery ticket hypothesis exist for BERT? With extensive experiments, their conclusion is: Only subnetworks found on the masked language modeling task (the same task used to pre-train the model) transfer universally; those found on other tasks transfer in a limited fashion if at all. This can be used to produce compressed BERT models and can have a broad impact in the NLP field. ========================== I have read the authors' response and appreciate their effort.

Strengths: 1. An interesting and important problem is studied. Especially, I believe "Transfer Learning for BERT Winning Tickets" is a very important problem, since this can be used to produce compressed BERT models for down-stream tasks. 2. The experimental details are clear and seem to be reproducible. 3. Several claims are examined and show positive results.

Weaknesses: 1. IMP is not a structured pruning method. Therefore, the claim of "it may be possible to reduce the cost (both energy costs and financial costs) of fine-tuning BERT models on downstream tasks" might be incorrect. It would be interesting to see if the lottery ticket hypothesis still holds for structured pruning of BERT.

Correctness: The empirical methodology is correct.

Clarity: The paper is well motived and well written.

Relation to Prior Work: The discussion is clear.

Reproducibility: Yes

Additional Feedback:


Review 2

Summary and Contributions: After reading the rebuttal: I've increased my score. 1) The authors corrected one of my main critiques -- I was incorrectly comparing their validation scores (using BERT-base) to the STILTs scores (using BERT-large). I have editing that part of my review so the other reviewers won't be concerned. 2) The updated Figure 2 is improved, and will help other readers understand. 3) It's great you report the average and standard deviation (the updated tables are improved). That's more than a lot of papers do. However, validation data is usually used to do model selection (i.e. select the best model) which is then evaluated on test data. Unfortunately there is a bit of a history of GLUE validation results being incorrectly compared (median of runs, mean of runs, max of runs) so to facilitate future comparisons I would recommend including in the appendix min / mean / max / std dev, or (if there are ~5 runs to get one average) just reporting the numbers themselves. 3.5) Updated Table 3 should include the Rewind results from the original paper, but with mean +/- std dev (Table 3 in the paper is just single numbers). 4) My complaint about the relaxed definition of a winning ticket isn't that it's *easier*, it's that you're redefining an already-defined term and then using it throughout your paper. I actually think that being within a standard deviation (or some other measure of being "close") is better for language problems, since there's often more variation in the results. But it's inappropriate to have one sentence near the beginning of the paper saying, "we define lottery tickets to be X" and then saying throughout the rest of the paper that you're finding lottery tickets. I would recommend you call what you're finding a "std dev lottery ticket" or something, since it's not a lottery ticket if it doesn't match the performance. ========== original review below ========== This paper analyzes the lottery ticket hypothesis for BERT trained on GLUE tasks. They show that they can find sparse networks (up to 90% sparse), and they claim to find some subnetworks which transfer to other tasks universally. After reading the rebuttal: They addressed some of my criticisms.

Strengths: They evaluate on the GLUE tasks and SQuAD, and they evaluate at different levels of sparsity, with varying amounts of rewinding. They also show performance of transferring subnetworks from each task to all other tasks, which would be a good experimental design if their baseline performance was good. Learning a lottery ticket on the masked language modeling task is quite nice -- it reinforces that MLM training can lead to good transfer performance. After reading the rebuttal: STILTs isn't a fair comparison (BERT-large vs BERT-base), but it would still be nice to see a more clear connection.

Weaknesses: The authors should use the standard evaluation metrics for each of the GLUE tasks. For example, MRPC is typically evaluated with the average of F1 and accuracy, but in this paper only accuracy is reported. The paper claims repeatedly that this is the first work to find lottery tickets for NLP tasks, and includes a definition of a winning ticket, but then relaxes the definition to be achieving performance within two points of the baseline. Either the authors can claim to find lottery tickets and use the definition of lottery ticket, or they can claim they can retrain models to within two points of the baseline, but not both. It's unclear why the baseline compared against is the average of three runs, and why that average performance is used e.g. in Table 3. Lottery tickets are from a single initialization -- this current setup means it's possible the authors took the best-performing model of three runs, trained lottery tickets on that, and compared against the average of the three runs (which would be unfair). To clear this up, the baselines used in Table 3 and beyond should be replaced with the performance of the single model that the lottery ticket is trained on. Figure 2 and Figure 3 don't actually show the performance, so it's unclear what the baseline is here. Figure 2 has coloring issues -- in the description of Figure 2 it's unclear what the number in each cell means, but presumably it is the improvement, so >0 should be blue and <0 should be yellow. If this is true (it seems consistent with Figure 3) then in Figure 2 there are many cells colored blue when they should be yellow, leading to it looking light there are many more successful transfers than there are.

Correctness: EDIT After reading rebuttal: I mistakenly was comparing their BERT-base results to BERT-large. I'm leaving it below in the "=" block so the other reviewers can see what the authors were addressing in their rebuttal. ============================================================ My main concern is that the baseline BERT performance is below others' reported performance, and the baseline tickets are evaluated against is the average of three runs. This could be cleared up by the authors, so it might not actually be a problem. It's not listed if the results are on validation or test, but either way, the baseline might be below what it should be, which would make finding subnetworks which achieve performance as strong as the baseline easier. For example, I've copied some BERT results from Sentence Encoders on STILTs: Supplementary Training on Intermediate Labeled-data Tasks below. Since it's unclear if the results here are test or validation, I've included the following format (excluding a few datasets because this submission doesn't use the standard evaluation metrics for them, though it should): Dataset: Submission_results vs (STILTs_validation or STILTs_test) MNLI: 82.4 +- 1.2 vs (86.2 or 86.3) QNLI: 89.1 +- 1.6 vs (89.4 or 91.1) MRPC: 85.3 +- 0.1 vs (89.0 or 85.4) RTE: 66.1 +- 4.4 vs (70.0 or 70.1) SST: 91.7 +- 1.1 vs (92.5 or 94.9) CoLA: 53.5 +- 1.3 vs (62.1 or 60.5) If these are validation results, MNLI, MRPC, and CoLA are likely too low. If these are test results, MNLI, SST, and CoLA are likely too low. RTE is on the edge, but variance for the RTE and CoLA are high in general. ============================================================= The results in this submission are described as the average across three runs, so maybe this is comparing single-run performance in STILTs vs the average of three runs? But if the reported number is the average of three runs, it's not clear that a single run of IMP should be compared to the average of three runs as a baseline.

Clarity: The bulleted lists of claims, which are then described in detail, are nice! Unfortunately some of the claims aren't clearly supported by the experiments, but the structure of the paper is good.

Relation to Prior Work: They claim this paper was done concurrently with Prasanna et al., 2020. I don't see that as a reason this should be rejected, they address it directly.

Reproducibility: No

Additional Feedback:


Review 3

Summary and Contributions: this paper proposes an unstructured pruning algorithm to find matching subnetworks (lottery tickets) in BERT and to be universal subnetwork which can also be transferrable to down stream task as well. To find the matching subnetwork, iterative matching pruning (IMP) is used, where a pre-trained model is fine-tuned for i number of iterations to find a subnetwork by pruning weights with small magnitude. Then this subnetwork is iteratively fine-tuned for (t-i) steps (by rewinding step i) to further prune small weights until reaching a sparsity goal. Finally the found matching subnetwork is initialized to the pre-trained weights (theta_{0}) and evaluated on downstream task to check wether it is a winning ticket.

Strengths: The paper examines different variations of weight pruning, such as random reinitialization, random pruning, to investigate the importance or proposed IMP pruning techniques. The ablation indicates that the suggested IMP can achieve highest sparsity for finding the winning ticket. It also indicates the necessity of pre-trained weights as starting point for finding winning tickets. - results show that a universal subnetwork is found on MLM task which can be transferrable to 10 down-stream task with 70% sparsity

Weaknesses: The proposed IMP pruning approach fails at finding matching subnetworks on task with relatively smaller training dataset. How this approach can be modified to address this problem?

Correctness: the empirical methodology is correct

Clarity: The results and discussion are well presented in the paper.

Relation to Prior Work: The related works are explained clearly.

Reproducibility: Yes

Additional Feedback: - How does the pruned attention looks like for the winning tickets? For example, for MLM task, does it learn to attend to more relevant words in the context? - There is a concurrent work [Prasanna et al.] which employs structured pruning algorithm. They did not examine transferability of subnetworks, and only study overlapping sparsity of subnetworks. There is no comparison to sparsity of matching subnetworks. -------------------------------------------------------------- Post Author Rebuttal: After reading the author response with new and modified experiments, I think this paper present a strong insight into winning tickets of nlp models. I am changing my score from 6 to 7.


Review 4

Summary and Contributions: The authors study the lottery ticket hypothesis in the context of modern NLP models with self-supervised pretraining, finetuning on numerous supervised NLP tasks, and potential transferability of pruned model structures across different supervised tasks.

Strengths: This is the most thorough experimental analysis of pruning of BERT/Transformer NLP models that I have seen so far. I like the +/- margin-of-error in Table 2. That is helpful. I also think it is likely that your Table 2 will be the "new baseline" that many researchers will use in future papers on neural network pruning and acceleration for natural language understanding tasks. I have personally been working on efficient neural nets for several years, and yet I have struggled with the question of "procedurally, where should I start if I want to apply the Lottery Ticket Hypothesis in my work?" The authors explain exactly how to apply the lottery ticket hypothesis to prune BERT. I am optimistic that this clear explanation will make pruning more accessible to more researchers, enabling someone to create a model that combines all the wins of novel model structure (e.g. MobileBERT), quantization (e.g. Q-BERT), pruning (Lottery Ticket Hypotheis), and efficient implementation of sparse networks (e.g. XNNPACK) into an extremely efficient NLP network that can be deployed on tiny devices. ---- Update after reading the rebuttal ---- Thanks to the authors for adding results to the paper for both metrics MNLI, QQP, STS-B, and MRPC. I am increasing my score from an 8 to a 9.

Weaknesses: In Table 2, are the GLUE results are from the test sets or the dev sets? I wasn't able to ascertain this from reading the paper; apologies if I have overlooked something. It would be helpful to see speedup results (on a hardware platform of your choosing) for the training and/or inference phases of your method. In the range of sparsity that you are targeting (70-90%), XNNPACK [1] has shown significant speedups over dense baselines on smartphone processors. [50] Erich Elsen, Marat Dukhan, Trevor Gale, Karen Simonyan. "Fast Sparse ConvNets." CVPR, 2019.

Correctness: Appears correct.

Clarity: Very well written.

Relation to Prior Work: Yes.

Reproducibility: Yes

Additional Feedback: - In the references, it might be helpful to add the arxiv identifier for preprints that are available on arxiv and do not yet have a journal or conference version. You do this well in reference [21], and I recommend doing the same for others such as [20], [22], [26]. (Chances are, you already plan to do this and just ran out of time before the submission deadline.) - Some of the GLUE tasks (QQP, STS-B, MRPC) are intended to be evaluated on two metrics (e.g. F1 and Accuracy). In Table 1 and 2, it mentions that you are only evaluating on one of the two metrics for these tasks. In the paper or the supplentary material, would it be possible to include a table similar to Table 2, but with results on both metrics for QQP, STS-B, MRPC? - It may make sense to remove the second 'the' from your title. The title would change from "The Lottery Ticket Hypothesis for the Pre-trained BERT Networks" to "The Lottery Ticket Hypothesis for Pre-trained BERT Networks." All that said, this is the best explanation and analysis of the Lottery Ticket Hypothesis that I have ever seen. Whether or note the paper is accepted to NeurIPS, I hope the authors will consider giving a tutorial on their work at a major conference. I think many will benefit from that.

[Author Response · NeurIPS 2020]

1 We thank the reviewers for their feedback, which we address point-by-point below. We include revised tables/figures at
2 the bottom, including: Table 2 with an updated winning ticket criterion (R2), Table 3 with two metrics for some tasks
3 (R2, R4), and Figure 2 split into two figures (R2). Due to limited space, reviewers must zoom in to see these items.

**Shared Comments** R2, R4: Some...tasks are intended to be evaluated on two metrics. Papers commonly report one or
both of two metrics for MNLI, QQP, STS-B, and MRPC. Table 3 shows both of these metrics for those tasks. Besides
STS-B (50% Pearson vs. 40% Spearman), winning ticket sparsities are the same on these tasks regardless of the metric.
R2, R4: Are the GLUE results from the test sets or dev sets? They are from the validation/dev sets.
R1: The claim of 'it may be possible to reduce the cost...of fine-tuning' might be incorrect. R4: It would be helpful to
see speedup results. We acknowledge that the real-world speedup of a sparse network depends on the software libraries
and the hardware. As R1 notes, the most direct way to do so is via structured sparsity. However, there is active work on
accelerating unstructured sparsity via software (Elsen et al. for CNNs as cited by R4) and hardware (sparse support on
the NVIDIA A100, GraphCore IPU, and Cerebras Wafer Scale Engine). These advances are a promising sign that future
work will be able to exploit our unstructured sparsity, and our results serve as a strong baseline to guide this research.
R1: LTH...for structured pruning of BERT R3: Comparison to the sparsity of matching subnetworks [in Prasanna et al.].
As we discuss in Section 2, Prasanna et al. study the LTH for BERT with structured pruning of entire attention heads.
We refer R1 to that section, where we discuss that work and other LTH results for structured pruning. Prasanna et al.
look at "subnetworks that achieve 90% of full performance" (less accurate than our winning ticket criterion) and report
how often each head survives pruning (rather than the overall sparsity), so we cannot directly compare to their results.

**Reviewer 1, Reviewer 4:** All technical comments addressed above. We acknowledge R4's presentation comments.

**Reviewer 2:** This paper overclaims on many things, so the claimed result should be taken with a grain of salt.
We have addressed all of the reviewer's concerns point-by-point below.
(A) This paper relaxes the definition [of a winning ticket] to be achieving performance within two points of the baseline.
Our motivation for the 2% threshold was to account for variation between runs (see (D)) rather than to introduce a more
permissive criterion. To make this clear, we have revised our winning ticket criterion to account for this variation in a
stricter, task-specific way. Specifically, we consider a subnetwork to be a winning ticket when the mean full BERT
performance is within one standard deviation of the mean subnetwork performance (computed over five runs; see (B)).
We updated Table 2 (below) accordingly; sparsities only change on STS-B (70% → 50%) and SQuAD (70% → 40%).
(B) Why the baseline compared against is the average of three runs, and why that average performance is used.
To clarify, we perform multiple runs with different random seeds for the data order; we report the average over these
runs. (We always use the same HuggingFace BERT initialization.) For each baseline run, we fine-tune with a random
data order. For each lottery ticket run, we train with a random data order, prune, rewind, and re-train with another
random data order. The updated Table 2 below shows means and standard deviations across five such runs. We average
over multiple runs in this way to show that our results are robust and are not cherry-picked. This is standard practice in
lottery ticket work [13, 14, 15, 16, 17, 18, 19], and it is required by the Machine Learning Reproducibility Checklist.
(C) Figure 2 has coloring issues. The colors and numbers in Figure 2 described two separate comparisons; we
acknowledge this was confusing, and we have split this into Figures 2a and 2b below. The numbers in Figure 2 showed
TRANSFER$(\mathcal{S}, \mathcal{T})$ minus the performance of unpruned BERT on task $\mathcal{T}$; this determined whether the transferred
subnetwork was a winning ticket. As we explain (L240), however, 70% sparsity is too sparse to find winning tickets on
several tasks. As such, we also compare whether transfer performance TRANSFER$(\mathcal{S}, \mathcal{T})$ is at least as high as same-task
performance TRANSFER$(\mathcal{T}, \mathcal{T})$ even if neither is a winning ticket. Cells in Figure 2 were blue when this was the case,
and this could be computed manually by checking if a cell's value was at least as high as the cell in the same-column
diagonal. We apologize for the confusion; we believe Figures 2a and 2b address this concern and improve clarity.
(D) BERT performance is below others' reported performance. We use the HuggingFace reference implementation of
BERT Base; our best numbers are in line with those reported by HuggingFace (see Tables 2 and 3 below). Reported
numbers can vary widely based on number of runs, metric (mean/median/best), and hyperparameter search [see *Show
Your Work*, Dodge et al. EMNLP 2019]. STILT is not comparable to our numbers: (1) It uses BERT Large, not BERT
Base. (2) It "perform[s] 20 random restarts...and report[s] the results...that performed best," while we report averages.

**Reviewer 3:** IMP...fails at finding matching subnetworks on tasks with relatively smaller training sets. IMP finds
winning tickets (which are a form of matching subnetwork) on all tasks, and "there is no discernible relationship
between the sparsities for each task and the properties of the task itself" (L164). Smaller training sets only seem to
affect rewinding, which we find to be unnecessary for our goal of uncovering sparse, transferable subnetworks.

Revised Table 2     Table 3: Both Metrics     Figure 2a: Transfer Winning Tickets     Figure 2b: Transfer vs. Same-Task

[Meta-Review · NeurIPS 2020]

All four knowledgeable referees support acceptance for the contributions because it gives answers to important problems that many people are interested in and gives insight to readers through many experiments in various environments. I also recommend acceptance. Please note the updated comments of reviewer R2 and consider revising your paper accordingly.